# Single-Nucleus and Spatial Transcriptomics Revealing Host Response Differences Triggered by Mutated Virus in Severe Dengue

**DOI:** 10.3390/v16111779

**Published:** 2024-11-15

**Authors:** Qian Chen, Yizhen Yuan, Fangzhou Cai, Zhe Li, Qiang Wei, Wei Wang

**Affiliations:** 1Institute of Laboratory Animal Sciences, Chinese Academy of Medical Sciences, Peking Union Medical College, Beijing 100021, China; qianchen1777@163.com (Q.C.);; 2National Center of Technology Innovation for Animal Model, State Key Laboratory of Respiratory Health and Multimorbidity, Key Laboratory of Pathogen Infection Prevention and Control (Peking Union Medical College), Ministry of Education, NHC Key Laboratory of Comparative Medicine, Institute of Laboratory Animal Science, CAMS & PUMC, Beijing 100021, China

**Keywords:** dengue virus, cumulative mutation, severe dengue, single-nucleus RNA sequencing, spatial transcriptomics, Nrg4

## Abstract

Dengue virus (DENV) infection causes various disease manifestations ranging from an asymptomatic state to severe, life-threatening dengue. Despite intensive research, the molecular mechanisms underlying the abnormal host responses and severe disease symptoms caused by evolved DENV strains is not fully understood. First, the spatial structure of mutant DENV was compared via in silico molecular modeling analysis. Second, employing single-nucleus and spatial RNA sequencing, we analyzed and verified transcriptome samples in uninfected, mild (NGC group), and severe (N10 group) liver tissues from murine models. In this study, we obtained a cumulatively mutated DENV-2 N10 with enhanced capability of replication and pathogenicity post 10 serial passages in *Ifnra^−/−^* mice. This variant caused severe damage in the liver, as compared with other organs. Furthermore, mutated DENV infection elicited stronger responses in hepatocytes. The critical host factor *Nrg4* was identified. It dominated mainly via the activation of the NRG/ErbB pathway in mice with severe symptoms. We report on evolved N10 viruses with changes observed in different organisms and tissue. This evolutionary variant results in high replicability, severe pathogenicity, and strong responses in murine. Moreover, the host responses may play a role by activating the NRG/ErbB signaling pathway. Our findings provide a realistic framework for defining disturbed host responses at the animal model level that might be one of the main causes of severe dengue and the potential application value.

## 1. Introduction

Dengue virus (DENV) is a single-stranded RNA virus transmitted by arthropods. After inoculation from the bite of an anthropophilic vector (*Aedes. aegypti* or *A. albopictus*), DENV can cause human clinical disorders ranging from almost asymptomatic dengue fever (DF) to severe, life-threatening dengue (SD) [1]. With the increase in international trade, travel, and climate change, dengue cases have risen over recent decades. Since the beginning of 2023, continuous transmission combined with an unexpected surge in dengue cases has led to more than 5,000,000 cases and over 5000 dengue-related deaths reported in more than 80 countries/territories and five WHO regions globally: south-east Asia, Africa, the western Pacific, the eastern Mediterranean regions, and the Americas [2,3,4,5,6,7,8,9,10].

Genetic variation and rapid evolution are common in RNA viruses due to the error-prone nature of the RNA-dependent RNA polymerase (RdRp) and viral replicase [11]. DENV is categorized into four distinct serotypes based on antigenic features [12]. Furthermore, each serotype is genetically sub-categorized into specific genotypes. Previous research has suggested that the serotype and/or genotype of DENV over time are associated with both the outbreak of epidemics and the severity of the disease [13]. Moreover, accumulating evidence indicates that DENV-2 contributes substantially to higher mortality in tropical and subtropical countries, compared to other DENV serotypes [14]. DENV-2 viruses from the cosmopolitan genotype clade caused unprecedented outbreaks in Sri Lanka in 2017 and Yunnan in 2015 [15,16], with unusual clinical disease manifestations including encephalitis, liver failure, and myocarditis [17].

Certain mutations have been suggested to provide DENV with a selective advantage, such as increased transmissibility, replication, or pathogenicity. A retrospective study of a DENV-2 outbreak in Cuba in 1997 suggested that a conservative single amino acid substitution (T164S) in non-structural protein 1 (NS1) was responsible for enhanced symptom severity among clinical cases as the epidemic progressed [18]. Other studies have shown that two tandem substitutions in the DENV-2 envelope protein (E)—namely, threonine to lysine at position 226 (T226K) and glycine to glutamic acid at position 228 (G228E)—promoted the infectivity of the virus in either mammalian hosts or mosquitoes [19]. Besides clinical studies, multiple lines of evidence from laboratory experiments with animals and cell lines support the association between specific DENV genetic mutations and disease phenotypes [20,21]. However, the host response to these continuously circulating and mutating viruses and the contribution of the cumulative mutations to severe disease in vivo remain to be understood.

In this study, we combined virological, pathological, and biochemical analyses; single-nucleus RNA sequencing; and spatial RNA sequencing to define how cumulative mutations in the DENV genome influence viral replication and pathogenicity in a mouse model. We showed that, compared to the prototypic New Guinea C strain of DENV-2, the cumulative mutant strain N10 results in severe hepatocyte damage and abnormal host responses. The NRG/ErbB signaling pathway was identified to possibly be involved in murine pathogenesis, which provides a potential target for research and improves our understanding of the potential correlations between viral mutations and host responses.

## 2. Materials and Methods

### 2.1. Viruses

DENV-2 strain New Guinea C (NGC) was purchased from ATCC (VR-1584™). DENV-2 N10 was obtained in our lab after 10 serial passages of NGC in *Ifnar^−/−^* mice. Briefly, mice were infected intravenously with NGC. At 3 days post-infection (dpi), infected mice were euthanized, and blood was harvested. Then, 100 µL of infected serum was intravenously delivered to a new cohort of *Ifnar^−/−^* mice. This process was repeated a further nine times to obtain the DENV-2 N10 isolate. Serum from each passage was stored at −80 °C for other tests. To produce stocks of the dengue virus for downstream experiments, three-day-old BALB/c suckling mice were inoculated intracranially with DENV-2 NGC or N10 (in a maximum volume of 20 µL), and the brain was harvested at 5 dpi and pooled in 10 mL of Dulbecco’s modified Eagle’s medium (DMEM) (11995065, Gibco, Waltham, MA, USA) and homogenized in M tubes using the RNA 01.01 program 1 for 1 min with gentleMACS Dissociator (130-093-235, Miltenyi, Bergisch Gladbach, NRW, Germany). The brain homogenate was collected and centrifuged at 1000× *g* for 10 min followed by filtration through 0.22 µm filters (SLGP033RB, Millipore, Burlington, MA, USA). The viral stocks were aliquoted (1 mL) and stored at −80 °C [22]. Titers were evaluated using the plaque assay with BHK21 cells [23]. All infection experiments involving DENV were performed in a Biosafety Level 2 (BSL-2) or Animal Biosafety Level 2 (ABSL-2) laboratory and authorized by the Institute of Laboratory Animal Sciences, Chinese Academy of Medical Sciences (ILAS, CAMS).

### 2.2. Cells

Vero cells and BHK21 cells were cultured with 5% CO_2_ at 37 °C in DMEM supplemented with heat-inactivated 10% fetal bovine serum (10099-141, Gibco, North Ryde, NSW, Australia) and 2% penicillin–streptomycin (15140122, Gibco, Grand Island, NY, USA). C6/36 cells were cultured with 5% CO_2_ at 28 °C in RPMI 1640 medium (11875119, Gibco, Grand Island, NY, USA) supplemented with 10% heat-inactivated fetal bovine serum and 2% penicillin–streptomycin.

### 2.3. Animals

C57BL/6 mice deficient in type I interferon receptors (*Ifnar^−/−^*) were obtained from Cyagen (Suzhou, China) Biotechnology Co., Ltd. The mice were bred and maintained in ABSL-2 at ILAS, CAMS. Four- to six-week-old mice (half male and half female) were used for the disease score experiments, viral load experiments, pathological experiments, and blood biochemical tests. A total of nine and three female mice were used in the single-nucleus RNA sequencing experiments and spatial transcriptome sequencing experiments, respectively. 

### 2.4. Animal Experiment

The mice were anesthetized via intraperitoneal injection of tribromoethanol (500 mg kg^−1^) and inoculated intravenously with 100 μL of DENV. Virus stocks were diluted to a final concentration of 10^6^ PFU/mL in serum-free DMEM. After infection, animals were weighed daily and monitored for 14 days. Clinical scores ranged from 1 to 5:1, slight loss of weight; 2, mild signs of lethargy; 3, hunched posture, fur ruffling; 4, hunched posture, ruffled and limited mobility, and increased lethargy; 5, moribund. Infected animals were euthanized directly when they lost >20% weight or >15% of their starting weight for three consecutive days. At 6 and 8 dpi, mice were anesthetized with tribromoethanol for the collection of blood and tissues (liver, spleen, kidney, brain, and intestine) that were stored at −80 °C until further processing.

### 2.5. RNA Quantification

Total RNA containing viral RNA and RNA transcripts corresponding to genes of interest was extracted using the QIAamp Viral RNA Mini Kit (Qiagen, 52906, Hilden, Germany) and RNeasy Mini Kit (Qiagen, 74106, Hilden, Germany). Then, the viral RNA was quantified with one-step TaqMan quantitative polymerase chain reaction (qPCR) (Qiagen, 204443, Hilden, Germany). For host gene transcripts, total RNA was reverse-transcribed into cDNA utilizing the Prime Script reverse transcription-polymerase chain reaction kit (Takara, RR037A, Kusatsu, Shiga, Japan). The host cDNA was quantified using SYBR Green qPCR (Takara, RR820A, Kusatsu, Shiga, Japan). The primer sequences are shown in Appendix A.

### 2.6. Histological Analysis

Animal tissues were collected and fixed in 10% paraformaldehyde for more than 24 h, followed by 70%, 80%, 95%, and 100% ethanol dehydration, paraffin embedding, and sectioning. A portion of slides was stained with hematoxylin and eosin (H&E). The remaining sections were used for immunohistochemical (IHC) staining. The steps are described as follows: H_2_O_2_ was added and incubated at room temperature for 10 min, and the tissue slices were placed in sodium citrate buffer at PH 6.0 for antigen repair. The sealing solution was added and incubated at room temperature for 60 min. The primary antibody (NS3 dengue virus protein: GeneTex GTX124252, Midland, TX, USA, diluted 1:500) was added and incubated at 4 °C overnight. Then, the slices were rinsed, and DAB was added as a chromogen. Finally, the ImageJ software (v.1.8.0) was used for NS3 protein quantification.

### 2.7. Multiplex Cytokine Analysis

Plasma collected from EDTA-anticoagulated blood was clarified via centrifugation. ProcartaPlex immunoassays with the Mouse Cytokine and Chemokine Panel 1A (EPX360-26092-901, Thermo Fisher Scientific, Waltham, MA, USA) were run according to the user guide. The samples were quantified on the Bio-Plex200 System (Luminex, Genk, Belgium).

### 2.8. Western Blotting Analysis

Liver tissues were weighed in a 1.5 mL Eppendorf tube and ground in a tissue lysate. Then, homogenized tissues were centrifuged at 4 °C and 12,000× *g* for 10 min to remove debris. Next, the protein concentration of the supernatant was determined using the bicinchoninic acid (BCA) method. The protein sample was denatured in loading buffer at 100 °C for 15 min and separated using 10% SDS-PAGE and then transferred to NC membranes. Membranes with transferred proteins were blocked with skim milk powder at room temperature for 1 h and then incubated overnight with the primary antibody solution at 4 °C. Primary antibodies against NRG4 (Abcam, ab174432, Cambridge, MA, USA, 1:1000), SULT1E1 (Santa Cruz, E-12, sc-376009, Dallas, CA, USA, 1:200) were used. The following day, the membranes were incubated with the secondary antibody for 1 h, and then the Immobilon ECL Ultra Western HRP substrate (Millipore, WBULS0100, Burlington, MA, USA) was utilized for visualization. Finally, ImageJ software was used for protein quantification.

### 2.9. Sequence Alignment

The complete genomic sequences of the NGC and N10 strains were assembled from sequences of PCR fragments amplified with Platinu Taq DNA Polymerase High Fidelity PCR (Invitrogen, 11304029, Carlsbad, CA, USA) and sequenced via Sanger sequencing (ABI 3730 capillary sequencer, Carlsbad, CA, USA). The primers for RT-PCR sequences are summarized in Appendix A. The nucleotide sequences were aligned with the ClustalW menu using the Bioedit editor. The mutant positions of amino acid sequences were compared and analyzed using the Molecular Evolutionary Genetics Analysis (MEGA) 11.0 software (version 11.0.13). Espript (http://espript.ibcp.fr/ESPript/ESPript/, accessed on 20 January 2020) online tools were used to compare several E and NS1 amino acid sequences from the NGC and N10 viruses. According to the homology modeling method, tertiary structures of the E and NS1 proteins of dengue strain N10 were predicted using the online tool Swiss-model (http://espript.ibcp.fr/ESPript/ESPript/, accessed on 20 January 2020). The E and NS1 proteins of the two strains were superimposed and compared using Pymol software (v.2.5.7).

### 2.10. 10× Genomics Chromium snRNA-Seq Data of the Mouse Liver

Fresh liver tissues were collected from *Ifnar^−/−^* mice uninfected and infected with NGC and N10 DENV-2 strains at 6 dpi (n = 3). Single-nucleus suspensions were generated according to the 10× Genomics Nuclei Isolation from Cell Suspensions Protocols [24]. The single-nucleus suspensions were loaded into microfluidic chromium chips with 3’ V3.1 chemistry and barcoded with a 10× Chromium Controller (10× Genomics). According to 10× Genomics single-nucleus RNA sequencing protocols, the Illumina Novaseq 6000 PE150 platform was then utilized to complete the library construction. Fastp was used to execute basic statistics on the quality of raw reads. The reads were mapped to the mouse genomes (mm10) via the 10× Genomics Cell Ranger pipeline (cellranger-6.0.0) using default parameters. We used snRNA-seq data from mouse hepatocytes in vivo for analysis. Cellranger aggr was used to aggregate output and to normalize, recompute, and analyze the combined data [25]. Then, we used the Seurat R package to reduce the dimensionality, cluster, and analyze the differential gene expression (DGE) [26]. Next, we used the clusterProfiler R package to implement GO and KEGG enrichment analysis [27]. Monocle 2 was used for pseudotime analysis [28]. Cellchat (version 1.6.1) was used for cell–cell communication analysis [29].

### 2.11. 10× Visium Spatial RNA Sequencing Library Preparation and Data Analysis

Fresh liver tissues from *Ifnar^−/−^* mice uninfected and infected with NGC and N10 DENV-2 strains at 6 dpi (n = 1) were concurrently frozen and embedded in optical cutting tissue (OCT) compound in liquid nitrogen. The RNA quality was assessed with an Agilent 2100 instrument. The Visium Spatial Tissue Optimization Slide and Reagent Kit (10× Genomics, PN-1000193) was employed to optimize permeabilization conditions, according to the Visium Spatial Tissue Optimization User Guide (CG000238, 10× Genomics). Visium spatial libraries were constructed using the Visium spatial library construction kit (10× Genomics, PN-1000184). Histology images and raw FASTQ files were processed with the Space Ranger (version spaceranger-2.0.0, 10× Genomics) software with default parameters [30]. The clusterProfiler R package was used to conduct the enrichment test for candidate gene sets based on hypergeometric distribution [27]. Pathways with corrected *p* values of less than 0.05 were considered as significantly enriched terms.

### 2.12. Statistics Analysis and Graphs

All the statistics were plotted and analyzed with GraphPad Software, version 9.4. Data are shown as the mean ± SEM. The log-rank (Mantel–Cox) test was used to analyze survival rates. Non-parametric Mann–Whitney and unpaired t-tests were used for the difference assessment of all other groups.

## 3. Results

### 3.1. The Evolution of DENV After Passing Through Ifnar1^−/−^ Mice Increases Viral Pathogenicity and Replicability

To identify DENV-2 mutations that are associated with severe disease in the *Ifnar1^−/−^* mouse model, we infected *Ifnar1^−/−^* mice at the age of 4–6 weeks intravenously (i.v.) with DENV-2 strain NGC (an international reference strain). Ten rounds of infection were performed to generate passed virus strains N1 to N10 (Figure 1A). Furthermore, the pathogenicity of these DENV strains was evaluated in *Ifnar1^−/−^* mice. At 3 days post-infection (dpi), most adaptive strains of DENV caused body weight loss in infected mice compared to the original NGC virus strain. A remarkable loss of body weight was observed in mice infected with the N6 and N8 virus strains (Figure 1B). At 6 dpi, starting from the sixth generation (N6), DENV strains were able to cause clinical symptoms in infected mice with a gradual increase in clinical scores thereafter. The ninth (N9) and tenth (N10) generation of DENV caused severe disease or even death in mice (Figure 1C,D). Thus, these findings suggest that DENV N10 increased disease pathogenicity and severity in mice.

The pathogenicity of the virus is closely related to viral replication in vivo [31]. We next sought to determine the replicability of passed DENV strains by detecting viral nucleic acid in the serum of infected mice at 3 dpi via a real-time quantitative PCR (qRT-PCR) assay. Our results showed that viral loads in serum samples of infected mice dramatically increased with successive generations. A significant difference in viral load between the original strain and passed strains began to appear from the sixth generation (N6 strain), which is consistent with the clinical score results. In N10 strain-infected mice, the highest viral load among all viral strains (the average value is greater than 6.68 × 10^4^ copies/μL) was detected and was 75 times that in NGC-infected mice (*p* < 0.0001) (Figure 1E), indicating that the enhanced pathogenicity of N10 in *Ifnar1^−/−^* mice was at least partly due to the rise in viral replication. To exclude the influence of host extracellular factors, we tested the level of viral replication in Vero cells and *Aedes albopictus* clone C6/36 cells in vitro. The results showed the magnitude of N10 infection in mammalian Vero cells and C6/36 cells was significantly greater than that of NGC at 4 and 5 dpi (Figure 1F,G).

### 3.2. Cumulative Mutations Occur in Adaptive DENV with Enhanced Replication and Pathogenicity

We sequenced the viral genomes and made alignments between the strain N10 and the prototypic NGC to investigate the genetic mutation of the viral genome post serial passages. We found a total of 11 non-synonymous mutations and two residue deletions in the N10 genomic coding sequence, which was mainly located in regions encoding the structural protein envelope (E) and non-structural proteins (NS1), including seven mutations in the *E* gene (S112G, I124S, K126E, E360G, I402F, I432V, and V484F) and two point mutations in the *NS1* gene (V177E, G235E). In addition, there was a mutation of E28G in *prM* and a deletion of A18 in *NS4A* (Appendix A). The E protein is the major determinant of dengue virus antigenicity and replicability [32,33], and NS1 is associated with DENV replication and pathogenesis [8,34]. To predict the secondary structure of the mutated virus and identify reliable sites of structural difference, we implemented an in silico molecular modeling analysis by comparing the spatial structure of the E and NS1 proteins between the NGC and N10 strains, respectively. We found that the two primary substitutions of isoleucine (Ile, I) with serine (Ser, S), a polar uncharged amino acid, at the 124th residue, and of glutamic acid (Glu, E) with glycine (Gly, G), resulted in a negative charge of the uncharged amino acid at the 360th residue (Appendix A). Mutations at the 124th residue have been reported to enhance the virulence of dengue virus [35]. Similarly, the prediction of the tertiary structure of the NS1 protein and the comparison of the tertiary structures between NGC and N10 showed that the main substitutions of the NS1 protein in N10 at the 177th residue (V177E) and the 235th residue (G235E) introduced negative charges (Appendix A). Notably, changes in the polarity and charge of an amino acid in the E or NS1 protein affect not only the tertiary structure of a protein but also its interactions with other pathogen recognition receptors [19]. Given the high pathogenicity and replicability of N10 described above, we concluded that this accumulation of non-synonymous mutations of the 124th and 360th residues of the E protein as well as the 177th and 235th residues of the NS1 protein in N10 may play a potential pathogenic role at key mutant sites in the mammalian host.

### 3.3. Cumulative Non-Synonymous Mutations in N10 Cause Severe Liver Damage in Mice

We next assessed the pathogenicity of a passed virus in different organs of infected mice. At 8 dpi, livers from NGC-infected mice showed infrequent infiltration of focal inflammatory cells. In contrast, livers in N10-infected mice showed moderate multifocal necrotizing hepatitis and vacuolar degeneration in hepatocytes (Figure 2A). No pathological change was observed in the spleens from NGC-infected mice. Spleens in N10-infected mice showed hyperplasia of red pulp cells (Figure 2B). The brains from NGC-infected mice showed meningeal vascular congestion and dilation, minimal infiltration of inflammatory cells, and mild glial cell hyperplasia in the cortex. Notably, brains from N10-infected mice showed not only microfocal hemorrhage but also vacuolar degeneration in the brainstem, inflammatory cell infiltration, and glial cell hyperplasia (Figure 2C). In kidneys and intestines, there was no obvious histopathogenic change in either NGC- or N10-infected mice (Figure 2D,E). Thus, the DENV N10 strain has different pathogenicity to different organs in *Ifnar^−/−^* mice. We further examined viral load in different organs infected with dengue virus. The results showed that the viral replication capacity of the N10 group was significantly higher than that of the NGC group, which was consistent with the results of the above in vitro experiments (Figure 2F). Then, we measured the circulating cytokine levels in the mice. The results showed that at 8 dpi, the levels of TNF-α, IL-17, and IL-23 in the N10 group were significantly increased compared with the uninfected group and the NGC group, indicating the increased pathogenicity of the N10 strain to the mice (Figure 2N).

As mentioned above, N10 strain infection has profound effects on the pathology of multiple organs. In the course of dengue disease, varying degrees of liver involvement are observed, such as elevated aminotransferase, liver enlargement, and acute liver failure [36,37,38]. The roles of the liver are to remove toxins from the body, secrete proteins, and regulate blood coagulation; these functions are closely related to dengue fever symptoms [39]. To verify the differential liver tropism of dengue virus, we examined the expression of non-structural protein 3 (NS3) at 6 dpi. The results showed that NS3 was mainly located in the necrotic area of hepatocytes in the N10 group, while a small amount of NS3 was located in endothelial cells in the NGC group (Figure 2G,H). Furthermore, we evaluated liver function in acute dengue infection using blood biochemical tests. At 6 dpi, levels of alanine aminotransferase (ALT) in N10-infected mice were significantly higher than those of NGC-infected or uninfected mice, with a marginal increase in AST (Figure 2I,J). Meanwhile, other liver function indicators, including total protein (TP), globulin (GLOB), and albumin (ALB) in N10-infected mice, were significantly lower than uninfected mice (Figure 2K–M). These results suggested that the N10 dengue strain with cumulative non-synonymous mutations causes liver damage in mice, especially hepatocyte damage.

### 3.4. N10 Strain Virus Induces Aberrant Host Responses in Mouse Livers

To further comprehensively explore host responses in liver tissue post N10 infection, we first performed single-nucleus RNA sequencing (snRNA-seq) on liver homogenates at 6 dpi (Figure 3A). After strict quality control, we obtained transcriptomes of 90,655 cells (uninfected control: 31,488; NGC: 39,167; N10: 29,000). The Uniform Manifold Approximation and Projection (UMAP) reduction technique revealed eight major cell clusters, including hepatocytes, Kupffer cells (KCs), endothelial cells (ECs), hepatic stellate cells (HSCs), monocyte-derived macrophages (MoMFs), T cells, B cells, and Rad51b+ cells, which were expressing different markers, respectively (Figure 3B,C). We were unable to identify cluster 16 with specific marker genes so this cluster was named “Rad51b+ cells” based on its highly variable genes (Figure 3B,C). The transcriptomics of hepatocytes, the primary undertakers of liver function, in NGC- and N10-infected mice was analyzed (false discovery rate (FDR)-adjusted *p* value < 0.05 and absolute log2 fold-change > 1). By comparing the differential expression genes (DEGs) of liver parenchymal cells in the NGC and N10 infection groups, it was found that there were significant differences in the expression levels of several genes, and the expression of these genes in the two groups of liver parenchymal cells was shown in the heatmap. The results indicated the upregulation of 32 DEGs and downregulation of 21 DEGs in N10-infected hepatocytes. The top 15 genes are shown (Figure 3D). Gene Ontology (GO) enrichment analysis showed that the sterol biosynthetic process, alpha-amino acid metabolic process, and cofactor binding the mitochondrial inner membrane and organelle inner membrane are the top biological functions in N10-infected hepatocytes (Figure 3E,F). Consistent with the results for histopathological analysis and the blood biochemical test, Kyoto Encyclopedia of Genes and Genomes (KEGG) enrichment analysis revealed that N10 infection caused the elevated expression of genes involved in non-alcoholic fatty liver disease and autophagy pathways. Additionally, genes associated with oxidative phosphorylation were significantly differentially expressed (Figure 3G,H). Collectively, these findings indicate that DENV N10 strain infection causes substantial aberrant host responses in the liver of the infected mice.

### 3.5. Trajectory of Hepatocellular Differentiation upon N10 Infection

Heterogeneity and molecular complexity are critical features of tissue damage. We further analyzed the pseudotime development of differential expression genes in hepatocytes between the NGC and N10 groups to predict differentiation states and time order by CytoTRACE to delineate the molecular characteristics of a host response and determine hub genes that regulate the biological process post NGC and N10 infections. The data first showed Uniform Manifold Approximation and Projection (UMAP) of hepatocytes and differentiation stages in the uninfected, NGC-, and N10-infected groups (Figure 4A,B, Appendix A). Next, we used Monocle2 to construct a biologically meaningful pseudotemporal path for hepatocyte ordering during DENV infection, which showed a clear and distinctive linear branch ordering hepatocyte clusters along pseudotime. This indicated that hepatocytes in the uninfected group were at the beginning of the time trajectory, while those of the infected groups, namely, NGC and N10, were at the end. There were two nodes in the hepatocytes’ pseudotemporal path (Figure 4C,D). This timeline was further mapped to the UMAP maps of the uninfected, NGC, and N10 groups (Figure 4B). Our results indicate the developmental status and trajectory of uninfected hepatocytes compared to infected hepatocytes.

We further analyzed the nodes to identify the changes in expression patterns during the progression of N10 infection. We found that each node had six different patterns of gene expression. Specifically, this analysis of branch node 1 revealed gene blocks including *Ddit4*, *Cidec*, *Serpina3c*, *Nrg4,* and so on (gene cluster 4–6), and gene blocks including *Shank2*, *Chka*, *C6*, and so on (gene cluster 1–3). Similarly, we also performed enrichment analysis for branch node 2. We found that common changed genes included *Ddit4*, *Serpina3c*, *Nrg4*, and so on (Figure 4E,F). Therefore, we further analyzed the expression patterns of these shared genes. Surprisingly, in the pseudotime analysis of DENV infection process, *Nrg4* was significantly different in the trajectory of NGC and N10 (Figure 4G,H, Appendix A).

These results showed that the expression levels of distinct genes were changed during dengue infection, which suggests that each gene may have a different order and timing to perform a function. We further explored the spatial characteristics of these genes.

### 3.6. Spatial Distribution of the Key Host Genes in the Process of N10 Infection

According to the expression characteristics of the above genes in the N10 group from pseudotime analysis, we explored the location distribution of these genes via high-resolution spatial transcriptomics (STs). Therefore, we used three sections of liver from uninfected control, NGC-, and N10-infected mice at 6 dpi for library preparation and sequencing. By filtering, annotating, and normalizing the raw data, we separately obtained expression data comprising 2058, 2267, and 2987 median genes per spot across 2401, 3970, and 3029 filtered spots on the ST arrays from uninfected, NGC, and N10 group mice. All data were subjected to downstream algorithm analysis (Figure 5A, Appendix A). Of note, only spots under the liver tissue sections were used for visualization and analysis. In terms of trajectory analysis, we ensured that the upregulated genes shared by the two branches included *Ddit4*, *Serpina3c*, *Nrg4*, *Sult1e1*, *Cxcl13*, and so on. Therefore, we used 10× Genomics Visium spatial transcriptomic sequencing to analyze the location and expression levels of these key genes to determine whether there were differences between the NGC and N10 groups.

As expected, we found that *Sult1e1* (Figure 5D,E) and *Cxcl13* (Figure 5H,I) were widely expressed in N10-infected liver tissues and expressed at higher levels than in the NGC group, the expression values of which were all over 2. *Nrg4* (Figure 5B,C) as well as *Ddit4* (Figure 5F,G) were also widely highly expressed in the N10 group, even though they were expressed at lower levels than *Sult1e1* and *Cxcl13*. These results suggested that these candidate genes, which are highly expressed in the N10 group, may play a role in abnormal host responses.

In summary, we identified some key genes that may play a responsive role in the host liver infected by the cumulatively mutated dengue virus strain N10. We used these genes as a basis to further identify the potential key signaling pathway in dengue infection.

### 3.7. Cell–Cell Communication in the Process of N10 Infection

Next, we sought to identify the underlying molecular mechanism of host response in severe dengue. Therefore, transcriptome profiles for each cell type from our snRNA-seq data (Figure 3B) were analyzed via CellChat. This analysis revealed 24 signaling pathways in the NGC group and 21 signaling pathways in the N10 group at 6 dpi. These mainly include the NRG, TGFb, EGP, and BMP pathways (Figure 6A,C, Appendix A). We conducted a network analysis of the NRG/ErbB signaling pathway between the NGC and N10 groups and identified that monocyte-derived macrophage (MoMF) populations are the most prominent signaling sources, and hepatocytes are the primary recipients for N10 infection. There was little difference in the other pathways (Figure 6B,C). In addition, *Nrg4* and *Erbb4* were involved in the NRG/ErbB signaling pathway of hepatocytes in response to DENV infection (Figure 6D, Appendix A). More importantly, during N10 infection, MoMFs as senders transmitted signals via Nrg2, which in turn triggered Nrg4 signaling in hepatocytes (Figure 6C,D).

### 3.8. Nrg4 Responds to the Accumulative Mutated N10 Dengue Virus Strain

To further validate our findings, the differential expression of *Cxcl13*, *Ddit4*, *Sult1e1*, and *Nrg4*, among others, was determined. As transcription precedes the translation of proteins, RNA was validated at 4 and 6 dpi, and protein was validated at 6 and 8 dpi (Figure 7A). The results revealed that the relative copy numbers of the above four key genes were markedly increased in N10-infected liver tissues, compared to those in the NGC control (Figure 7B–E). At the protein level, SULT1E1 was markedly elevated at 6 dpi of N10 infection (Figure 7F,I). In addition, NRG4 in the N10 group was significantly higher than that in the NGC group at 8 dpi (Figure 7G,H). Collectively, these data indicated that Sult1e1 and Nrg4 are potential host response factors for DENV N10 strain infection. In summary, NRG4 may act through the NRG signaling pathway in SD.

In conclusion, investigation of the potential hub genes that respond to cumulatively mutated dengue viruses in vivo may offer insights into the treatment of severe dengue, which is important in the research and development of potential drug targets for dengue viruses.

## 4. Discussion

DENV is an arthropod-borne virus of the genus *Flavivirus* in the family *Flaviviridae* [1]. Studies on the antigenic and genetic evolution indicate that the average mutation rate of dengue virus is around 7.5 × 10^−4^ mutations/position/year, which is only slower than influenza virus and HIV [40]. Moreover, mutations resulting from the long-term circulation of dengue virus across the human population also drive viral gene evolution [41]. While tremendous progress has been made in the field of virus mutation and evolution, the researchers have only studied the effects of viral mutation on humans unilaterally in terms of virology and immunology. The mutation and evolution of viruses and host response to these distinct virus strains are fundamentally intertwined [42]. The mutation of antigens leads to an alternation in the host response to viral infection [43]. It has been demonstrated that the serial passage of DENV in host cells and mice results in an increase in inflammation and mortality in mice. However, few studies have explored the relationship between cumulative mutations of dengue virus and host responses.

This study presents vital evidence of phenotype changes caused by N10 infection in the mouse model. First, we obtained a DENV strain, N10, with cumulative mutations by mimicking the way viruses evolve in response to host immune pressure. Next, through the combination of two omics approaches, we analyzed the association of virus mutation with the transcription and translation of host genes in response to viral infection and revealed the molecular characteristics and spatial and temporal distribution of host responses. In this study, the combination of cumulative genetic mutational analysis with newly established single-nucleus RNA sequencing and spatial transcriptome sequencing laid the foundation for us to investigate effects of viral mutation and evolution on host response in the liver.

Neuregulin 4 (NRG4) is a member of the neuregulin family that acts on the EGFR receptor family. In our study, mutant DENV strain N10 resulted in the significantly enhanced expression of NRG4 at both the transcript and protein levels in bulk liver tissue. Computational analysis suggested that, in the NGC-infected liver, only NRG4 and Erbb4 signals were transmitted in hepatocytes. However, in the N10-infected liver, NRG2 derived from MoMFs also transmitted signals through the NRG/ErbB pathway. We propose that future experiments studying this cell–cell communication may help us to better understand the relationship between viral mutations and changes in host responses.

However, there are still shortcomings in this study. Although the mutations (I124S, K126E, E360G, I402F, and I432V) in the E protein of DENV have been observed in passed viral strains of this study in animals and reported in clinical isolates, it is important to further explore the specific mutations associated with pathogenicity via reverse genetics technology [44,45,46,47,48,49,50,51,52,53]. In addition, NRG4 has been shown to play an important role in liver disease in DENV-infected mice, and it will be of great significance to further investigate the effect of the evolution of the viral strain in a human population on changes in gene expression in clinical patients with severe dengue disease.

Together, the results reported here provide unique insights into host responses to DENV N10 infection in the liver and the identification of hub genes during this infection process. This study allowed for a more comprehensive understanding of the effect of virus mutation on host response in the liver. In conclusion, these data provide important information for the development of host response to DENV infection.

## Figures and Tables

**Figure 1 viruses-16-01779-f001:**
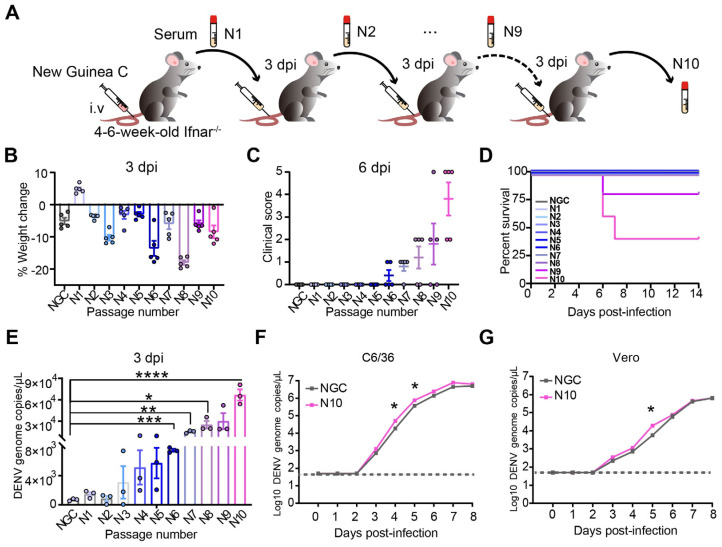
The enhanced pathogenicity and viral replication in *Ifnar1^−/−^* mice infected with successively passed DENV strains. (**A**) Graphical representation of the method of generating DENV-2 N10 virus. *Ifnar^−/−^* mice were infected intravenously with 10^6^ PFU/mL of prototypic DENV-2 NGC (n = 3). The infected mice were euthanized at 3 dpi, and serum was separated and inoculated intravenously into a new cohort of mice. This process was repeated 10 times in total. (**B**–**D**) Percentage of weight change (**B**), clinical score (**C**), and survival rate of infected mice (**D**). (**E**) Viral loads in serum at 3 dpi from infected mice were detected via qRT–PCR. (**F**,**G**) Cells were infected with NGC or N10 at a multiplicity of infection (MOI) of 0.1. The copies of viral genomic RNA of DENV2-NGC and N10 in C6/36 (**F**) and Vero cells (**G**) measured with qRT-PCR from 0 to 8 dpi. * *p* < 0.05, ** *p* < 0.01, *** *p* < 0.001, and **** *p* < 0.0001.

**Figure 2 viruses-16-01779-f002:**
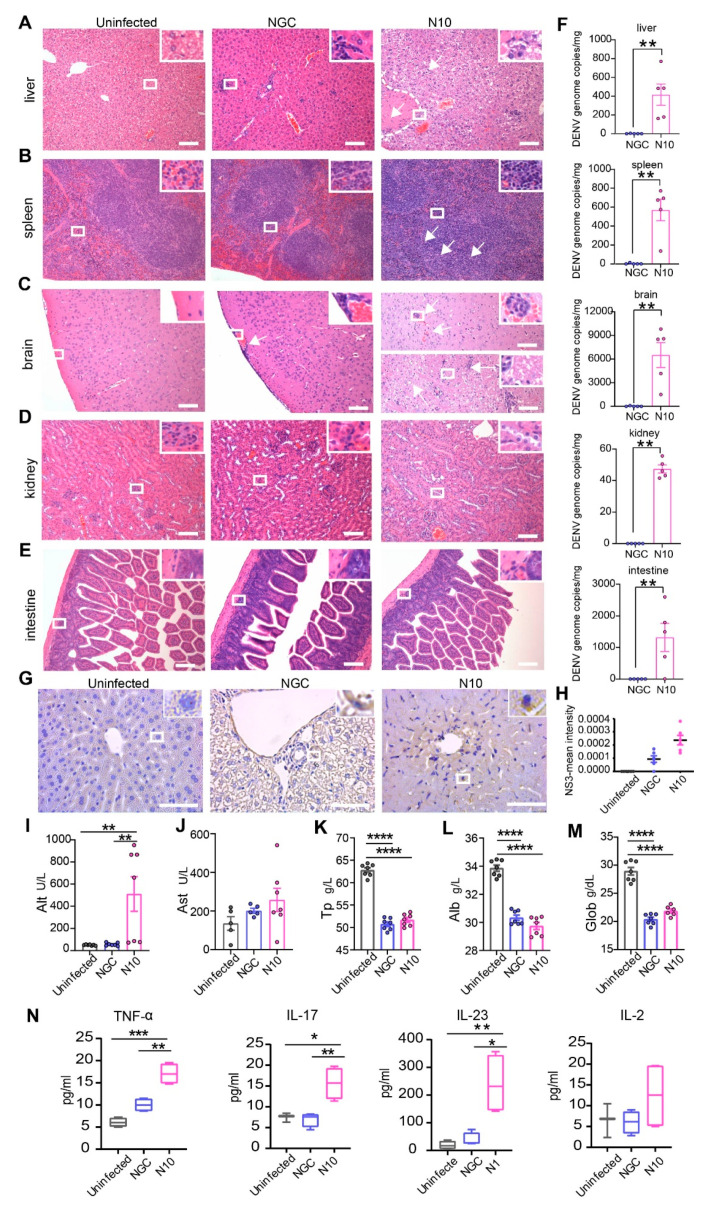
Dengue virus with cumulative mutations causes pathological changes. *Ifnar^−/−^* mice were infected intravenously with 10^6^ PFU/mL of DENV-2 NGC and N10 (n = 5). (**A**–**E**) Histopathology of the liver (**A**), spleen (**B**), brain (**C**), kidney (**D**), and intestine (**E**) from the uninfected, NGC-, and N10-infected mice at 8 dpi. The white arrows in the images showing H&E staining all represent the pathological changes mentioned in the results section. Histological images are representative of no fewer than five mice (scale bars, 50 µm). (**F**) Viral load results for liver, spleen, brain, kidney, and intestine. (**G**,**H**) Localization of the DENV NS3 in the liver (**G**) and quantitative results (**H**). (**I**–**M**) Blood biochemical analysis of uninfected, NGC-, and N10-infected mice at 6 dpi, including Alt (**I**), Ast (**J**), total protein (TP) (**K**), albumin (ALB) (**L**), and globulin (Glob) (**M**). Circulating cytokine analysis of uninfected, NGC-, and N10-infected mice at 8 dpi, including TNF-α, IL-17, IL-23, and IL-2 (**N**). * *p* < 0.05, ** *p* < 0.01, *** *p* < 0.001, and **** *p* < 0.0001.

**Figure 3 viruses-16-01779-f003:**
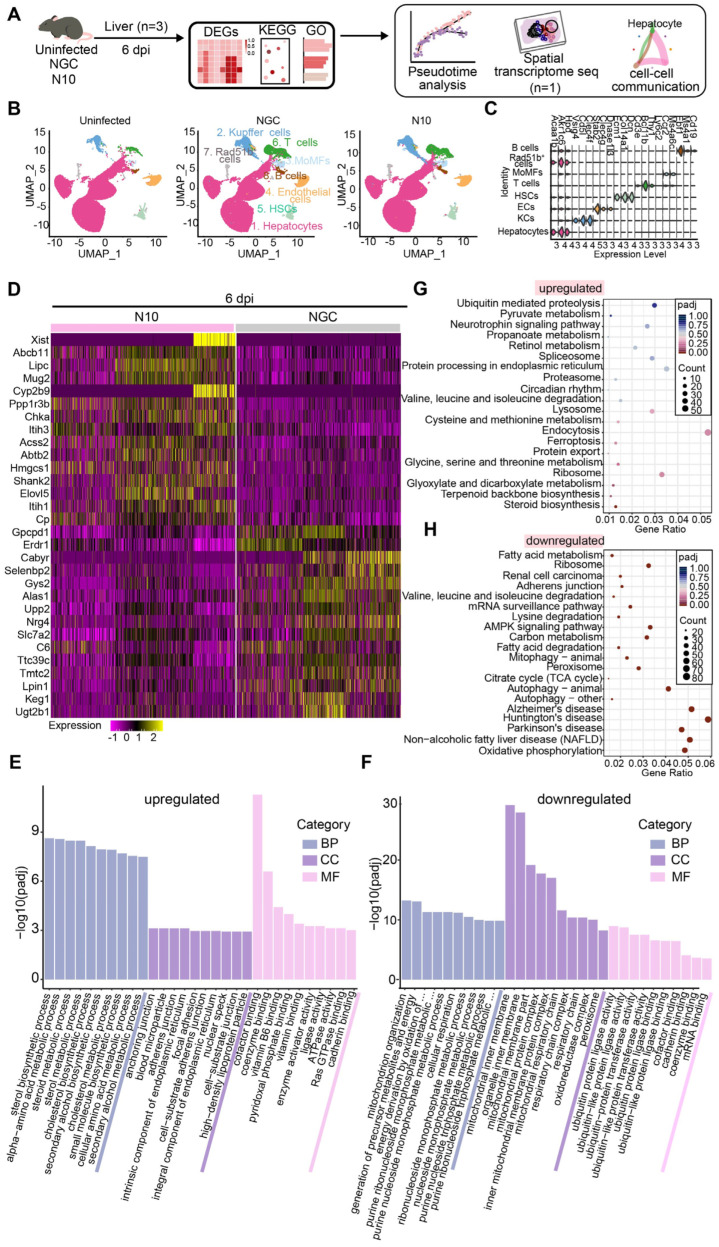
DENV strain N10 induces aberrant host responses in the liver. (**A**) Flow diagram of bioinformatics analysis. (**B**) The distribution of cell clusters in the livers from uninfected, NGC-, and N10- infected mice was shown in UMAP chart. (**C**) The distribution of typical cellular marker expression for different clusters was shown in violin plot. (**D**) Identified DEGs, (**E**,**F**) GO, and (**G**,**H**) KEGG upregulation and downregulation analysis in hepatocyte populations between the NGC and N10 groups. Dot sizes represent the proportion of all genes from GO analysis that were considered as markedly differentially expressed (gene ratio). Negative log10 FDR-adjusted *p* values related to each pathway are plotted.

**Figure 4 viruses-16-01779-f004:**
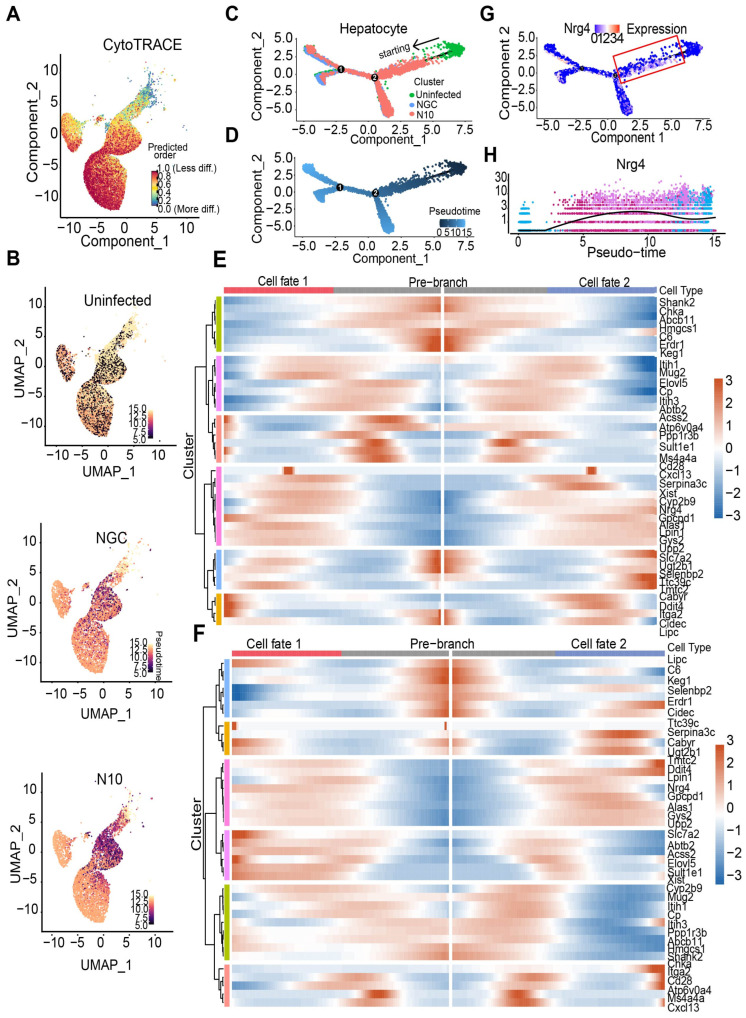
Pseudotime analysis of hepatocyte genes in response to NGC or N10 dengue virus. (**A**) UMAP of hepatocytes by differentiation state inferred using CytoTRACE. (**B**) Pseudotime trajectory projected onto the UMAP of hepatocytes. (**C**,**D**) Pseudotime analysis of the uninfected, NGC-infected, and N10-infected hepatocytes. The trajectory spans uninfected (arrow) to NGC-infected to N10-infected hepatocytes. (**E**,**F**) Heatmaps showing different gene expression patterns during the differentiation of node 1 and node 2 over pseudotime. (**G**,**H**) The gene expression distribution of *Nrg4* over pseudotime.

**Figure 5 viruses-16-01779-f005:**
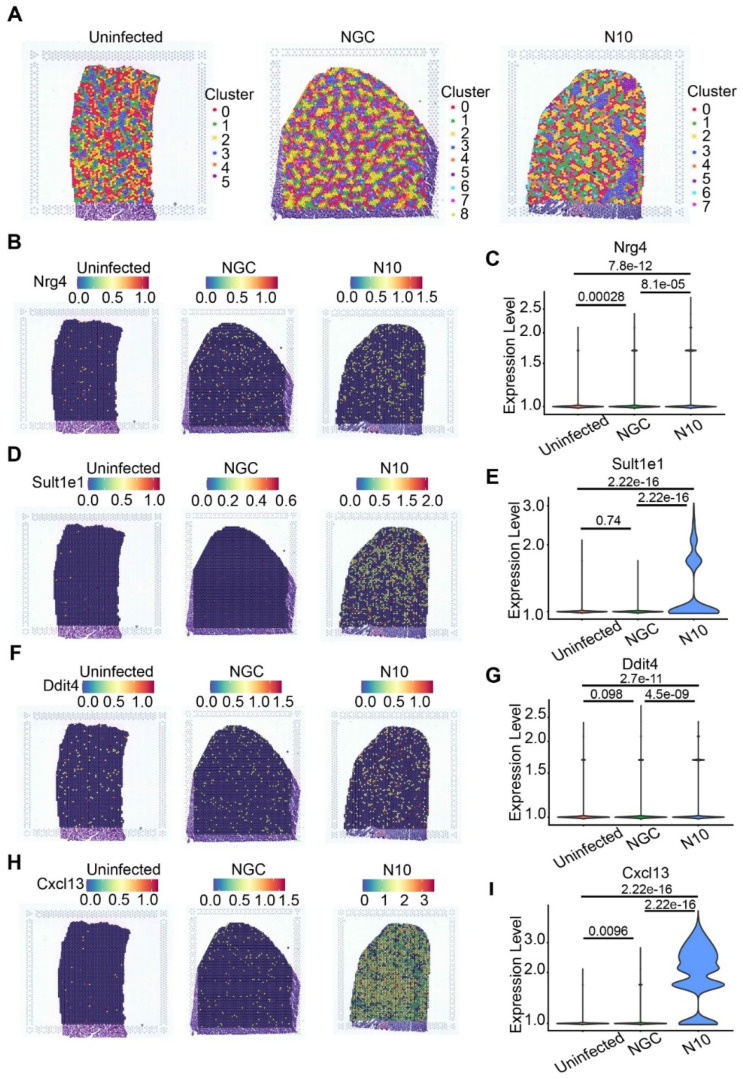
Spatial distribution analysis of hepatocyte genes in response to NGC or N10 dengue virus. (**A**) Spatial location of clusters in the uninfected group, NGC group, and N10 group. (**B**–**I**) Spatial expression distribution and expression quantity of candidate genes in the uninfected, NGC, and N10 groups. Candidate genes include Nrg4 (**B**,**C**), Sult1e1 (**D**,**E**), Ddit4 (**F**,**G**), Cxcl13 (**H**,**I**).

**Figure 6 viruses-16-01779-f006:**
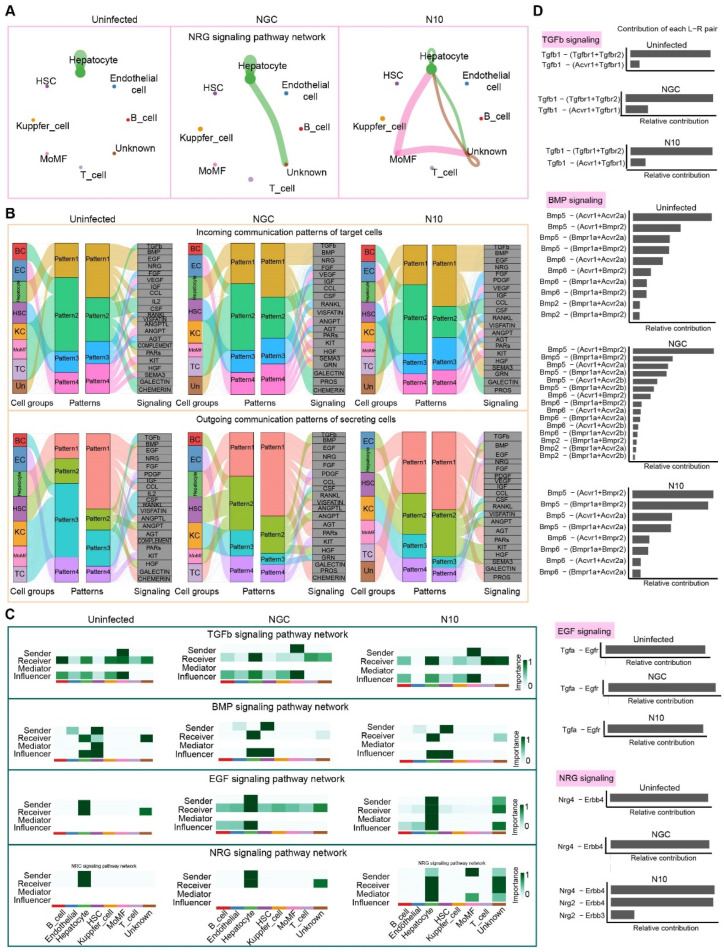
Cell–cell communication in dengue infection. (**A**) Involvement in cellular interactions of the NRG signaling pathway. (**B**) The outgoing (**bottom**) and incoming (**top**) communication patterns of different cells. (**C**) The heatmap shows the relative importance of different cell types (senders, receivers, mediators, and influencers) based on the computed network centrality measures of TGFb, BMP, EGF, and NRG signaling, respectively. (**D**) Relative contribution of ligand–receptor pairs.

**Figure 7 viruses-16-01779-f007:**
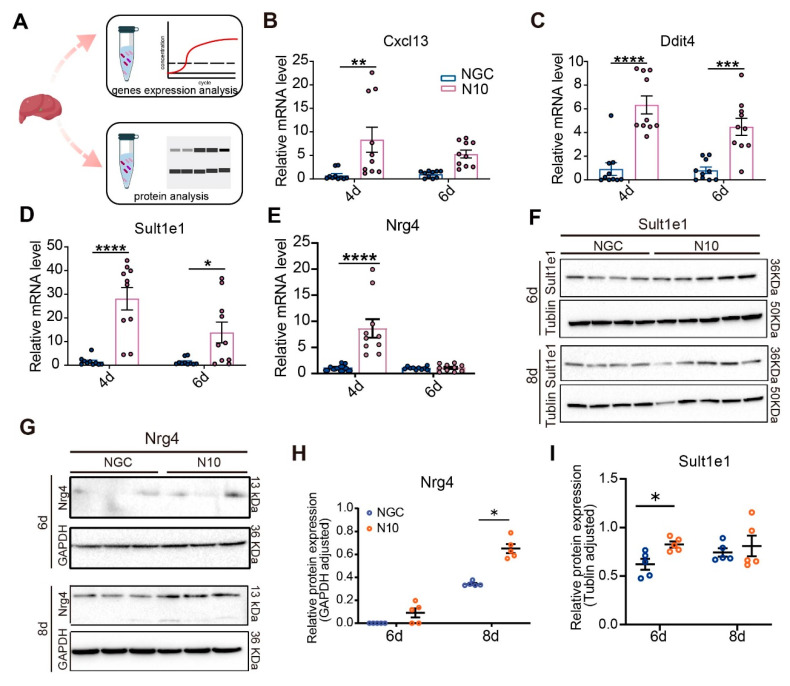
N10 dengue virus infection triggers change in *Nrg4*. (**A**) An overview of molecular validation (**B**–**E**). The relative mRNA expression levels of Cxcl13 (**B**), Ddit4 (**C**), Sult1e1 (**D**), and Nrg4 (**E**) in the liver tissues (n = 10). (**F**–**I**) The relative protein expression levels of SULT1E1 (**F**,**I**) and NRG4 (**G**,**H**) in the liver tissues (n  =  5). * *p* < 0.05, ** *p* < 0.01, *** *p* < 0.001, and **** *p* < 0.0001.

## Data Availability

Single-nucleus RNA sequencing datasets generated in this project are available from the Gene Expression Omnibus (GEO; accession number GSE250505). The complete genome sequence of the N10 stain in this study has been deposited at GenBank (OR838780). Data use requires authorization from the corresponding author.

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
