# Peer review of "Single-Nucleus and Spatial Transcriptomics Revealing Host Response Differences Triggered by Mutated Virus in Severe Dengue"

_viruses, 2024, doi:10.3390/v16111779_

Round 1

Reviewer 1 Report

Comments and Suggestions for Authors

Title: Single-nucleus and spatial transcriptomics revealing host re-sponse differences triggered by mutated virus in severe dengue

Chen et al, Viruses, 2024

This is an interesting study using novel, complex and new technologies to provide a level of detail that is not routinely undertaken in studying the host-virus interplay. The study has focused on the liver in the context of DENV infection in an IFNAR-/- mouse using NGC strain, and serially passaged isolates that have increased capacity to induce disease. While this is certainly interesting data, appears to be technically sound, the paper is not well described and can be improved in terms of both clarity of language and scientific descriptions. There are some important points of inaccuracy or needing clarification. Importantly, the discussion is lacking – with only the biology of NRG4 discussed, and no further discussion on any of the mutations, or any other outcomes of the significant data from the omics and spatial analysis. Further specific comments below:

General text edits, inaccuracies and queries

·       ‘rapidly evolving dengue strains’ – what is the evidence for this?

·       Introduction: Not sure that severe dengue fever (SDF) is the recommended terminology – should just be dengue fever, or severe dengue

·       NS1 is not the viral protease (page 2, second paragraph)

·       Liver involvement is not the most important clinical sign for DENV.

·       The text does not step the reader through these complex figures very well and could be improved to facilitate critical interpretation by the readership

Methods, details lacking

·       No indication of RT step in methods for PCR

·       Dengue virus NS3 protein antibody (GeneTex. GTX124252) = incomplete detail

·       Western blot detail incomplete – how was signal detected?

·       Animals: ‘anesthetized with bromethol’ – incomplete details?

Queries on results and interpretations

·       Fig1. How was virus normalised to compare N1 and N10 and NGC?

·       Pathogenicity measurements could be improved to reflect known DENV pathology in humans and mice eg. vascular leak syndrome measurement? Circulating cytokines (eg.TNF-a), GIT bleeding? - no change in gut was seen in this study, please comment on how this mouse model relates to these other measures used in other mouse models.

·       Fig 2 – DENV expressed as ‘copies per ul’? normalised to tissue RNA?

·       Introduction does not set the scene with specifics of the gaps in knowledge that rationalise the study and could be improve

·       Single nucleus and spatial RNA analysis – more detail, in what cells, from in vivo or in vitro???

·       Figure 1 – please add in day for clinical score and weight loss assessment (panels B and C). I am surprised that NGC is not already pathogenic in these IFNAR-/- mice, as NGC is already a passaged isolate from mouse brain? Panel F and G are not convincingly different over time, and statistically different only at a few selected time points. Please comment

·       DENV N10 isolate = a plaque purified isolate? This is relevant to mutations seen – are these all on the one virus genome. I would expect a quasispecies of virus in the serum after 3 days of infection. Were the strains plaque purified before sequencing to assess sequence of individual total viral genomes and suggest these changes are linked on one viral genome? How does N10 compare to other described mouse adapted DENV strains?

·       Fig 2 could be improved with some arrows on the images to highlight the pathologies described in the text, especially for a virology audience of the journal readership

·       At what time point was snRNAseq undertaken ? – point of liver dysfunction or pre-empting disease? – speaks to interpretation of roles of these responses in driving pathology or responding to dysfunction

·       Section 3.5 is unclear what was done experimentally (RNAseq analysis over time?) compared to computationally. Please clarify, including Figure 4

·       The spatial transcriptomics analysis is interesting, but the description could be improved for clarity. It would be beneficial to also link spatial analysis to the presence of the virus in the liver and if any histology can be colocalised such as the presence of veins/arteries, with foci of infiltrating cells or macrophages (or kuppfer cells), which could then be linked to outcomes in 3.7

·       Section 3.8 states analysis at 4 and 6 dpi, but then refers to mRNA changes at 8 dpi – please clarify time points used for analysis

·       a caveat for looking at virus-host responses is that this study was undertaken in the context of IFNAR-/- mice. What do you think these strains might do in the context of immunocompetent mice?

Comments on the Quality of English Language

Some aspects are well written, but clarity can be improved (scientifically), while overview for English language would also be of benefit.

Author Response

Thank you very much for the patient and meticulous comments of the reviewers. We will upload the replies to the reviewers' comments in pdf form. Please see the attachment.

Reviewer 2 Report

Comments and Suggestions for Authors

Viruses 2977713

Critique

Chen, et al. explore Dengue virus (DENV) infection in an interferon receptor knock-out (IFNAR-/-) mouse model. DENV serotype 2 was serially passaged 10 times in IFNAR-/- mice.  The authors characterized the genetic changes, clinical disease, pathology, virology.  In addition, they assessed host responses in liver using state-of-the-art transcriptional analyses (snRNA-seq and spatial transcriptomics). Although an immunocompromised mouse model has limitations in recapitulating disease that occurs in human patients, I believe the work is presented well and it will be of interest to the field. I recommend it for publication.

Minor edits:

There are a few minor edits to consider.

1.     Figure 1, panel E.  Possibly change the scale on the y-axis so that the values of NGC can be seen.  I assume it was not at “0” as it appears here.

2.     Figure 4, panels E and F.  Possibly enlarge these panels even a little so that the gene names are more readable.

Author Response

Thank you very much for the patient and meticulous comments of the reviewers. We will upload the replies to the reviewers' comments in word form.Please see the attachment.

Reviewer 3 Report

Comments and Suggestions for Authors

Dengue virus (DENV) is a major global health problem, with a heterogeneous range of clinical outcomes. Chen et al. derived a mutant DENV-2 (N10) by serially passaging a common lab isolate (NGC) in Ifnar-deficient mice, and found that N10 displayed increased viral replication and worse clinical outcomes. To uncover the molecular basis behind this phenotypic difference, they profiled gene expression of cells from mice infected with these two viruses, and hypothesized that NRG/ErbB signaling is a key factor differentially regulated between these two viruses. I was specifically asked to comment on the genomics involved in this manuscript. The overall approach seems reasonable to me, and this is potentially an interesting set of data (depending on the generalizability). However, there is a lot of technical detail lacking that makes it hard to interpret how the data is actually being analyzed. The validation of the NRG/ErbB signaling and, in particular, the generalizability of N10 to clinical DENV cases could also be improved.

1. Technical detail lacking for genomics

1a. From page 10: "The transcriptomics of hepatocytes, the primary undertakers of liver function, in NGC- and N10-infected mice was analyzed". Does this mean that the authors subsetted their data just on hepatocytes, and did not include non-parenchymal cells or infiltrating immune cells? Those are potentially valuable sources of information.

1b. Fig. 3 at least appears to be from solely hepatocytes, but there's no indication of subclustering of hepatocytes, no UMAP or other descriptive figure of the overall dataset, so it's hard to evaluate data and cell quality. If there are multiple 'cell fates' as inferred by the pseudotime analysis, then presumably there should be multiple hepatocyte cell states, and you would expect that the cells would be unevenly distributed between these states among the different infection conditions. One could perform differential gene expression between the subclusters, which is a simpler and more interpretable method than pseudotime analysis.

1c. Columns in Fig. 3B are not labeled. It's unclear if they are individual cells or clusters, and also unclear how many animals are included and whether there is animal-to-animal variability.

1d. It's not clear how CellChat was run. The schematic in 3A is confusing, as it appears that the Visium data was used for CellChat. However, Visium doesn't provide single-cell resolution, so it's not clear where molecular signatures of monocyte-derived macrophages (MoMFs) are coming from. Are they derived from the snRNA-seq instead? This is not clearly described in the text or the figures, see also point 1a.

1e. It's not clear to me how the pseudotime trajectory is being calculated. Is this semi-supervised or unsupervised? If semi-supervised, then the authors should describe what features were used to generate the ordering (was it some molecular signature of hepatocyte differentiation? Where did it come from?). If unsupervised, then it's not clear to me how the authors are able to create a directionality. Does it even really make sense to have a pseudotime ordering with the different conditions overlapped in this case? Presumably, not all the cells are infected with N10, and so it might be more logical to create an ordering just for the N10-infected condition, where some cells are uninfected (an origin, so t=0), and then infected cells would be further along to some infected cell state. The authors might have to attempt to look for DENV reads in their data, which might be difficult since this is snRNA-seq and not scRNA-seq (viral RNA would be expected to be exclusively in the cytoplasm).

1f. Was any kind of cell type deconvolution attempted from the spatial transcriptomics data? It may help to tease apart effects from different cell types, as N10-infected livers will have many more immune cells than uninfected or NGC-infected livers, which will confound Visium data.

2. Validation.

2a. Technical validation. The authors hypothesize that disease is correlated with monocyte-derived macrophage (MoMF) signaling to hepatocytes. They formulated this hypothesis using single-cell methods, and therefore should perform validation with single-cell, non-genomics methods (flow cytometry, immunohistochemistry (IHC), immunofluorescence (IF), etc), rather than bulk RT-qPCR or bulk WB. Traditionally, since multiple slices of a tissue are taken per sample, spatial transcriptomics is meant to be paired with IHC or IF of the adjacent tissue slices, so that researchers can identify both cell type and gene expression simultaneously.

2b. Biological validation. N10 is a mutant DENV which has never before been observed in patients. In order to generalize this finding between N10 and other cases of severe DENV, the authors can cross-reference their data with existing publicly available patient data which is abundant for comparisons of severe vs mild DENV. Without this comparison, it's difficult to say that N10 meaningfully reflects what actually occurs in patients, which limits the excitement generated by the wonderful data provided by this manuscript.

Round 2

Reviewer 3 Report

Comments and Suggestions for Authors

Mostly resolved issues

1a. From page 10: "The transcriptomics of hepatocytes, the primary undertakers of liver function, in NGC- and N10-infected mice was analyzed". Does this mean that the authors subsetted their data just on hepatocytes, and did not include non-parenchymal cells or infiltrating immune cells? Those are potentially valuable sources of information.

Authors' reply: We appreciate for your valuable comments. We showed that, compared to the prototypic New Guinea C strain of DENV 2, the cumulative mutant strain N10 results in severe hepatocyte damage and abnormal host responses. Combined with the clinical studies and autopsy results of patients with severe dengue, severe dengue infection can lead to a large number of liver cell necrosis. Hepatocyte necrosis may affect liver function, so we wanted to further investigate the effects of severe dengue on the host hepatocyte response. However, we also agree with you very much, so we have explored in depth the effects and interactions of non parenchymal cells or infiltrating immune cells on liver cells in other stories. At the same time, we supplemented the UMAP from the livers of uninfected, NGC and N10 infected mice to clarify the changes in cell types before and after DENV infection (Fig 3B, 3C).

Reviewer reply: Looks great, thank you for clarifying in the figure and in the text.

1b. Fig. 3 at least appears to be from solely hepatocytes, but there's no indication of subclustering of hepatocytes, no UMAP or other descriptive figure of the overall dataset, so it's hard to evaluate data and cell quality. If there are multiple 'cell fates' as inferred by the pseudotime analysis, then presumably there should be multiple hepatocyte cell states, and you would expect that the cells would be unevenly distributed between these states among the different infection conditions. One could perform differential gene expression between the subclusters, which is a simpler and more interpretable method than pseudotime analysis.

Authors' reply: Thanks for your comments. As mentioned above, we supplemented the UMAP of the overall dataset (Fig 3B, 3C). In this study, we focused on the dynamic changes of the gene expression profile of liver parenchymal cells during dengue virus infection. If different cell states are directly compared, the dynamic change trend of some genes may be ignored. Therefore, we directly adopted pseudotime analysis to obtain the evolutionary sequence of liver parenchymal cells after infection, so as to describe the dynamic change of gene expression profile under the influence of virus.

Reviewer reply: Looks great, thank you for clarifying in the figure and in the text.

1c. Columns in Fig. 3B are not labeled. It's unclear if they are individual cells or clusters, and also unclear how many animals are included and whether there is animal-to-animal variability.

Authors' reply: 1.1 Thank you very much for your careful review. We added relevant information. “By comparing the differential expression genes (DEGs) of liver parenchymal cells in NGC and N10 infection groups, it was found that there were significant differences in the expression levels of several genes, and the expression of these genes in the two groups of liver parenchymal cells was shown in the heatmap.”(Results 3.4 N10 strain virus induces aberrant host responses in mouse livers). 1.2 In this study, liver tissues of three mice were sequenced in each group. (Materials and Methods 2.10. 10×Genomics Chromium snRNA-seq data of the mouse liver). 1.3 This study used Ifnar-/- mice, C57BL/6 mouse inbred strain, a common strain used to construct animal models of dengue virus infection. It has the characteristics of homozygous gene, stable heredity, consistent phenotype and clear background data and the differences between the mice were almost negligible.

Reviewer reply: Looks great, thank you for clarifying.

1d. It's not clear how CellChat was run. The schematic in 3A is confusing, as it appears that the Visium data was used for CellChat. However, Visium doesn't provide single-cell resolution, so it's not clear where molecular signatures of monocyte-derived macrophages (MoMFs) are coming from. Are they derived from the snRNA-seq instead? This is not clearly described in the text or the figures, see also point 1a.

Authors' reply: As described above, we supplemented the UMAP of the overall dataset (Fig 3B, 3C). And we are sorry for the confusing schematic. We have revised the flow chart according to your comments (Fig 3A). We used snRNA-seq data for CellChat analysis.

Reviewer reply: The new figure is much clearer, thank you. Please also revise the text in section 3.7 (page 16) to reflect the use of snRNA-seq, something like: "Therefore, transcriptome profiles for each cell type from our snRNA-seq data (Fig. 3B) were analyzed via CellChat".

2b. Biological validation. N10 is a mutant DENV which has never before been observed in patients. In order to generalize this finding between N10 and other cases of severe DENV, the authors can cross-reference their data with existing publicly available patient data which is abundant for comparisons of severe vs mild DENV. Without this comparison, it's difficult to say that N10 meaningfully reflects what actually occurs in patients, which limits the excitement generated by the wonderful data provided by this manuscript.

Authors' reply: Thank you for your significant reminding. We thoroughly compared the reports of mutations consistent with strain N10 in animal models, clinical and epidemiological literature. We thoroughly compared the reports of mutations consistent with strain N10 in animal models, clinical and epidemiological literature. And the same mutation sites (I124S, K126E, E360G, I402F) are marked, which may have certain warning significance (Discussion) Some important references can be seen below.

Reviewer reply: Looks great, thank you for clarifying in the Abstract.

Unresolved: pseudotime validation (1e, 1f, and 2a)

1e. It's not clear to me how the pseudotime trajectory is being calculated. Is this semi-supervised or unsupervised? If semi-supervised, then the authors should describe what features were used to generate the ordering (was it some molecular signature of hepatocyte differentiation? Where did it come from?). If unsupervised, then it's not clear to me how the authors are able to create a directionality. Does it even really make sense to have a pseudotime ordering with the different conditions overlapped in this case? Presumably, not all the cells are infected with N10, and so it might be more logical to create an ordering just for the N10-infected condition, where some cells are uninfected (an origin, so t=0), and then infected cells would be further along to some infected cell state. The authors might have to attempt to look for DENV reads in their data, which might be difficult since this is snRNA-seq and not scRNA-seq (viral RNA would be expected to be exclusively in the cytoplasm).

Reply: Thank you for your professional comments. We used unsupervised computation, which evolves from uninfected to infected states according to evolutionary starting points set by cell biological significance. Then, because DENV did not have poly A tail and as you mentioned, we sequenced liver tissue samples using snRNA seq method, it was difficult to detect viruses in the cytoplasm, it could not be detected. Therefore, in this paper, we focused on the evolution of the global gene expression profile of hepatocellular parenchymal cells in the case of infection, and therefore set the hepatocellular parenchymal cells in the uninfected group as the starting point of evolution. The left branch of the infected group (NGC and N10), which coincides with the left side of the uninfected group, may be uninfected liver cells. (Fig 4C-D; Supplementary Fig 2B).

1f. Was any kind of cell type deconvolution attempted from the spatial transcriptomics data? It may help to tease apart effects from different cell types, as N10-infected livers will have many more immune cells than uninfected or NGC-infected livers, which will confound Visium data.

Authors' reply: Thanks for your detailed comments, in this study we directly focused on the overall expression distribution of genes of interest. However, we very much agree with your point of view, and we will continue to pay attention to the distribution of cell types in the liver of DENV-infected mice.

2a. Technical validation. The authors hypothesize that disease is correlated with monocyte-derived macrophage (MoMF) signaling to hepatocytes. They formulated this hypothesis using single-cell methods, and therefore should perform validation with single-cell, non-genomics methods (flow cytometry, immunohistochemistry (IHC), immunofluorescence (IF), etc), rather than bulk RT-qPCR or bulk WB. Traditionally, since multiple slices of a tissue are taken per sample, spatial transcriptomics is meant to be paired with IHC or IF of the adjacent tissue slices, so that researchers can identify both cell type and gene expression simultaneously.

Authors' reply: Thank you for your suggestion. The main objective of this study was to investigate the effect of mutated DENV infection on the host liver response. And this study used spatial transcriptomics (ST) to analyze the distribution of interest genes in high throughput, and then screened them by bulk RNA and bulk WB, finally identifying the up-regulated expression of targeted genes in the damaged liver caused by DENV infection. ST can better reflect the overall change of gene expression profile.

Reviewer reply (let's call this point 3): I'm still not convinced that pseudotime analysis is really the best approach here. What is the biological interpretation of pseudotime in this dataset? It can't be time = 0 is uninfected, and time = 15 is a very infected cell, because uninfected cells are located both at time = 0 and time = 15. Pseudotime can be useful in some instances, but when unsupervised, it's difficult to interpret the biology of what it's actually trying to measure. I think the best approach here is to take all of the snRNA-seq profiles from all 3 conditions, perform subclustering, then plot UMAPs and just try to see if there are multiple cell states. Ideally one subcluster has only cells from the infected conditions, and ideally that cluster is also high in expression of Nrg4 and various other genes. I think that's the simplest way to visualize your data and prove the point that the manuscript is trying to make. If that is included in the manuscript, I would have no further concerns on this point.

Reviewer reply point 4: Even with the subclustering UMAPs, the other issue is that there's still no single-cell validation. The authors bring up spatial transcriptomics in their reply, but the particular technique that they use, Visium, is not a single-cell method. The resolution is 55 microns, on the order of 5–10 cells merged together. One alternate hypothesis that explains their data—bulk WB, bulk RNA (RT-qPCR), and Visium spatial transcriptomics—is that N10-infected livers have the highest amount of infiltrating immune cells, and the immune cells (not hepatocytes) are actually the cell type that predominantly expresses Cxcl13, Ddit4, Sult1e1, Nrg4, etc. If the authors do not wish to perform further experiments (IF/IHC, etc), then I think they should add a caveat to the Discussion section (page 20). I suggest changing to "In our study, mutant DENV strain N10 resulted in the significantly enhanced expression of Nrg4 at both the transcript and protein levels in bulk liver tissue. Computational analysis suggested that, in the NGC-infected liver, only NRG4 and Erbb4 signals were transmitted in hepatocytes. However, in the N10-infected liver, NRG2 derived from MoMFs also transmitted signals through the NRG/ErbB pathway. We propose that future experiments studying this cell-cell communication may help us to better understand the relationship between viral mutations and changes in host responses."
